# Polymeric Composites with Silver (I) Cyanoximates Inhibit Biofilm Formation of Gram-Positive and Gram-Negative Bacteria

**DOI:** 10.3390/polym11061018

**Published:** 2019-06-09

**Authors:** S. R. Lotlikar, E. Gallaway, T. Grant, S. Popis, M. Whited, M. Guragain, R. Rogers, S. Hamilton, N. G. Gerasimchuk, M. A. Patrauchan

**Affiliations:** 1Department of Microbiology and Molecular Genetics, Oklahoma State University, Stillwater, OK 74078, USA; Shalaka.l@gmail.com (S.R.L.); erin.gallaway@okstate.edu (E.G.); treyon@ostatemail.okstate.edu (T.G.); manita.guragain@usda.gov (M.G.); rendirogers@gmail.com (R.R.); jsarahd@okstate.edu (S.H.); 2Department of Chemistry, Missouri State University, Springfield, MO 65897, USA; Snow87@live.missouristate.edu (S.P.); ger.nick@live.com (M.W.)

**Keywords:** Biofilm, antimicrobial, Pseudomonas aeruginosa, Staphylococcus aureus, Silver (I) coordination polymers, cyanoximes, thermal stability, polymeric composites, mechanical strength, density and porosity, leaching studies

## Abstract

Biofilms are surface-associated microbial communities known for their increased resistance to antimicrobials and host factors. This resistance introduces a critical clinical challenge, particularly in cases associated with implants increasing the predisposition for bacterial infections. Preventing such infections requires the development of novel antimicrobials or compounds that enhance bactericidal effect of currently available antibiotics. We have synthesized and characterized twelve novel silver(I) cyanoximates designated as Ag(ACO), Ag(BCO), Ag(CCO), Ag(ECO), Ag(PiCO), Ag(PICO) (yellow and red polymorphs), Ag(BIHCO), Ag(BIMCO), Ag(BOCO), Ag(BTCO), Ag(MCO) and Ag(PiPCO). The compounds exhibit a remarkable resistance to high intensity visible light, UV radiation and heat and have poor solubility in water. All these compounds can be well incorporated into the light-curable acrylate polymeric composites that are currently used as dental fillers or adhesives of indwelling medical devices. A range of dry weight % from 0.5 to 5.0 of the compounds was tested in this study. To study the potential of these compounds in preventing planktonic and biofilm growth of bacteria, we selected two human pathogens (Gram-negative *Pseudomonas aeruginosa* and Gram-positive *Staphylococcus aureus*) and Gram-positive environmental isolate *Bacillus aryabhattai*. Both planktonic and biofilm growth was abolished completely in the presence of 0.5% to 5% of the compounds. The most efficient inhibition was shown by Ag(PiCO), Ag(BIHCO) and Ag(BTCO). The inhibition of biofilm growth by Ag(PiCO)-yellow was confirmed by scanning electron microscopy (SEM). Application of Ag(BTCO) and Ag(PiCO)-red in combination with tobramycin, the antibiotic commonly used to treat *P. aeruginosa* infections, showed a significant synergistic effect. Finally, the inhibitory effect lasted for at least 120 h in *P. aeruginosa* and 36 h in *S. aureus* and *B. aryabhattai*. Overall, several silver(I) cyanoximates complexes efficiently prevent biofilm development of both Gram-negative and Gram-positive bacteria and present a particularly significant potential for applications against *P. aeruginosa* infections.

## 1. Introduction

Healthcare-associated infections are becoming increasingly prevalent in the United States. According to the 2014 report from the Center for Disease and Prevention (CDC), the estimated burden of nosocomial infections was 722,000 cases leading to about 75,000 fatalities. A significant number of such infections are commonly associated with implanting surgical devices. These infections are increasingly difficult to treat due to continuously growing prevalence of antibiotic resistant bacteria in hospital settings and lead to considerable morbidity and healthcare costs [1]. Pathogenic bacteria usually enter the surgical site during operative procedures or infect the wounds post-surgically [2]. Such surgical implant infections are caused by diverse species of bacteria, including species of *Staphylococcus, Streptococcus* and *Pseudomonas* genera [3]. *Staphylococcus* is considered the leading etiologic agent of implant-associated infections [4]. Device-related infections caused by *S. aureus* and *S. epidermidis* account for 50–60% of the cases [5]. *P. aeruginosa* represents a particular challenge due its antibiotic resistance (reviewed in Reference [6]) and ability to colonize and form biofilms on diverse surfaces, including those of devices, such as cardiac pacemakers, defibrillators and prosthetic cardiac valves [7]. *Pseudomonas* infections occur in about 4% of the estimated one million patients receiving implants each year [5]. In 2017, the World Health Organization (WHO) has listed *P. aeruginosa* and *S. aureus* among priority pathogens that require development of new antibiotics (WHO news release, 2017).

The development of implant-associated infections commonly involves adhesion of bacterial pathogens to the surface of an implant and formation of biofilms. The National Institutes of Health (NIH) reported that among all bacterial and chronic bacterial infections, 65% and 80%, respectively, are associated with biofilm formation. Biofilms are microbial communities that grow on surfaces. They are usually embedded into self-produced extracellular polymeric matrices, including polysaccharides, proteins and DNA. Bacterial biofilms are known for increased resistance to antibiotics, antimicrobials and host immune responses [8,9]. The increased biofilm resistance introduces a particularly critical clinical challenge in case of implant-associated infections, since managing such infections is difficult due to requirement of prolonged antibiotic therapy. In case this therapy fails, surgical interventions to remove the infected devices are required, which are risky for patient health condition and costly [10,11]. Therefore, there is a critical need in better alternatives and the development of novel infection-prevention strategies and antibacterial agents [3].

Application of metals, including silver (Ag), for treating human diseases is a vigorously expanding area [12,13]. From the time of Hippocrates, Ag has been known to exhibit a broad spectrum of antibacterial properties [14]. Strong bacteriostatic and bactericidal effects of Ag are generally explained by its interactions with bacterial DNA and inactivation of bacterial respiration enzymes [15]. In addition, interaction with Ag ions may cause generation of reactive oxygen species [16,17,18]. Disturbance of membrane permeability and inhibition of transcription and translations have also been reported [19,20]. There is also evidence that silver nanoparticles (AgNPs) directly interact with the cell surface, penetrating the cell membranes and causing leakage and eventually cell death [21,22]. Due to the increasing prevalence of multidrug resistant bacteria, applications of Ag as an antimicrobial have become of profound interest, particularly considering low toxicity of Ag to human cells and tissues [23]. Recent studies showed that Ag exhibits antibacterial activity against both planktonic (free swimming) and biofilm bacteria [24,25,26]. Silver reduces infection rates when used to coat surfaces of various implants, such as urinary and central venous catheters [27,28,29] and ventilator endotracheal tubes [30]. However, despite a strong potential of Ag for clinical applications, there are several challenges. The biggest challenge for most Ag(I) inorganic compounds with antimicrobial properties is a reduction of metal cation to elemental Ag(0) upon exposure to light and heat [31,32]. Other challenges include the chemical reactivity of Ag(I) compounds with unsaturated organic compounds.

For a number of years, we have been investigating Ag complexes with cyanoximes—a new class of the oxime-based versatile ampolydentate ligands [33,34] with the general formula NC-C(N=OH)-R, where R is electron withdrawing group [35,36]. The main purpose of this ongoing work is to study the effect of combination of the aforementioned biological properties of Ag with a well-established spectrum of biological activities of cyanoximes. The latter include growth-regulating [37,38] and antimicrobial [39,40,41] properties. We initiated synthesis and characterization of structures and properties of the new family of silver cyanoximates (reviewed in Reference [42]) and performed a preliminary assessment of their antimicrobial activity [43], followed by their further detailed characterization [44]. In the most recent work, we obtained two previously unknown AgL (L = 2-oximino-2-cyano-N-morpholylacetamide, MCO- and 2-oximino-2cyano-N-piperidineacetamide, PiPCO-), which were added to the list of visible light and heat stable, poorly water soluble antimicrobial compounds [45]. Also, we pioneered the addition of solid powders of Ag(MCO) and Ag(PiPCO) into the light-curable acrylate-based polymeric composite, widely used in medical practice [45]. This enables applications of these compounds as antimicrobial additives to adhesives and fillers for indwelling medical devices. No such studies were conducted for other previously reported silver(I) cyanoximates.

Here, we aimed a comprehensive study of the largest series of twelve Ag(I) cyanoximates so far, which previously showed visible light and thermal stability, poor water solubility and a potential for antimicrobial activity. Our objectives were to:(1)improve high-yield chemical synthesis of a series of Ag(I) cyanoximates designated as Ag(ACO), Ag(BCO), Ag(CCO), Ag(ECO), Ag(BIHCO), Ag(BIMCO), Ag(BOCO), Ag(BTCO), Ag(MCO), Ag(PiPCO) and Ag(PiCO) yellow and red polymorphs designated as Ag(PICO)-yellow and Ag(PICO)-red (Scheme 1);(2)perform mixing of solid Ag compounds by homogenizing in flowable light-curable acrylate polymeric composite and carry out photo-polymerization reactions to obtain new solid materials;(3)study thermal properties, morphology, of the newly obtained solid polymers using the TG/DSC and SEM methods as well as investigate their density, porosity and mechanical strength;(4)characterize leaching of Ag(I) ions from the new solid polymers into solutions;(5)examine antibacterial efficacy of the formed solid polymers against biofilm-forming pathogenic *P. aeruginosa*, *S. aureus* and environmental *B. aryabhattai* bacteria.

## 2. Experimental: Materials and Methods

### 2.1. Chemicals and Instrumentation

All solvents and basic chemicals, such as AgNO_3_ (≥99.0%), were reagent grade (Sigma Aldrich, St. Louis, MO, USA) and used without further purification. The cyanoximes used in this study (Scheme 1) were obtained in 2–5 g quantities according to reported procedures [43]. Identification and purity of ligands were confirmed by the elemental analyses on C, H, N content (Atlantic Microlab, Norcross, GA, USA), TLC chromatography on silicagel Al-backed plates with UV-indicator (Merk) and melting points measurements (Digimelt apparatus, without correction).

Chemical components of dental composites were obtained from Sigma Aldrich (St. Louis, MO, USA): triethylene glycol dimethacrylate (TEGMA), diurethane dimethacrylate (UDMA), Bisphenol A glycerolate dimethacrylate glycerol/phenol 1(Bis-GMA), silicagel (SiO_2_), as inert filler, hollow, non-porous glass beads, camphorquinone (CPQ) 97%, ethyl 4-(dimethylamino) benzoate (EDBMA). More specific information regarding these materials is shown in Appendix A

Non-halogenated buffer solutions used in the leaching experiments were Trimethylol Aminomethane (TRIS-Base ≥99.8%; 0.5 M) and 3-(4-Morpholino)propane sulfonic acid (MOPS) with a useful buffering range of pH 7–9 and pH 6.5–7.9, respectively [46]. Both solid chemicals were provided by Fisher BioReagents and both solutions were prepared according to standard laboratory procedures in DDI-water, to a pH of 7.

Brain Heart Infusion broth (BHI) and Tryptic soy broth (TSB) were purchased from Teknova (Hollister, CA, USA).

### 2.2. Inductively Coupled Plasma (ICP) Analysis on Ag Content

ICP analyses were carried by using the ICP-AES spectrometer (Varian Liberty, Palo Alto, CA, USA) at wavelength of 328.068 nm for detection of Silver(I) ion. An instrument search window was set to 0.080 nm and 0.120 nm with the scan in either direction. Standards for ICP method were obtained from Ricca (PAG1KN-100, Silver ICP Standard, 1 mL = 1 mg of Ag as 1000 ppm stock solution of AgNO_3_ in 3% Nitric Acid; Trace Element Analysis Grade, Ricca, Arlington, TX, USA). Standard solutions for Silver(I) ions determination were calibrated at 50.00 ppm, 10.00 ppm, 5.00 ppm, 1.00 ppm, 0.10 ppm concentrations in acidified double deionized (DDI) water. The latter was obtained by adding dropwise 1 M nitric acid into 1 L DDI water to pH 5.5.

### 2.3. Spectroscopy

The identity of the synthesized AgL complexes (L = cyanoximate anions from Scheme 1) was confirmed by: (1) recording their IR-spectra (Bruker FT-ATR instrument, Billerica, MA, USA; 500–4000 cm^−1^ range with 4 cm^−1^ resolution) and (2) recording their diffusion-reflectance spectra (Cary 100 Bio equipped with PMT Labsphere integrating sphere; 350–800 nm range) and matching those with the previously reported spectra [44,47]. Both types of spectra were obtained at room temperature.

### 2.4. Thermal Analysis (TA)

Solid samples of pure AgL in amounts of 6–30 mg were used for the thermogravimetric/ differential scanning calorimetry (TG/DSC) measurements using the Q-600 thermal analyzer from TA Instruments (TA, New Castle, DE, USA) (Appendix A). Complexes were heated under atmosphere of pure N2 from 30 to 1400 °C with 10°/min rate in a T-ramping mode. Data were processed with the aid of TA Universal Analysis software package and presented in Appendix A.

### 2.5. Optical Microscopy (OM)

Photographic images of the synthesized AgL complexes were obtained by using the Motic X300 microscope (Motic, Richmond, BC, Canada) at ×20 magnification and were used for the assessment of homogeneity of these microcrystalline solids (Appendix A).

### 2.6. Synthesis of Ag(I) Cyanoximates

Preparation of twelve Silver(I) complexes of 1:1 composition that include cyanoxime anions was accomplished using a modified straightforward one-pot two steps reaction [42,43,44,47,48]. Thus, the deprotonated cyanoxime, obtained upon the addition of K_2_CO_3_ to the HL (Scheme 1) forming bright-yellow solution in water, received an aqueous solution of AgNO_3_, which was added dropwise at room temperature and intense stirring: (a) 2 HL + K_2_CO_3_ = 2 KL + H_2_O + CO_2_ ↑ and (b) KL + AgNO_3_ = AgL ↓ + KNO_3_.

Preparations of Ag(CCO), Ag(BIHCO) and two polymorphs of Ag(PiCO) – red and yellow – were never previously reported and therefore, presented below.

### 2.7. Silver(I) α-Oximido-(2-pivaloyl)acetonitrile, Ag(PiCO), Red Form

To 10 mL solution of K(PiCO) at room temperature, prepared by addition of 0.112 g of K_2_CO_3_ to 0.250 g of HPiCO with the aid of a sonicator, a stoichiometric amount of AgNO_3_ (0.275 g) was added dropwise within a minute under stirring. Fibrous salmon-pink precipitate immediately forms. After ~10 min stirring the precipitate was filtered, washed with 5 mL of water and dried in a desiccator over H_2_SO_4_(c). Yield: 95%; compound rapidly decomposes at ~150 °C. Analytical data: for AgC7H9N2O2 calculated, (found): C – 32.21 (31.96); H – 3.48 (3.41); N – 10.73 (10.67).

### 2.8. Silver(I) α -Oximido-(2-pivaloyl)acetonitrile, Ag(PiCO), Yellow Form

Actual amounts of ingredients were the same as in previous preparation. To a hot (at ~90 °C) aqueous solution of in situ deprotonated with K_2_CO_3_ PiCO- in 10 mL of water a stoichiometric amount of AgNO_3_ was slowly added in the dark. Upon addition of a drop of silver nitrate solution turbidity of Ag(PiCO) complex appears but slowly dissolves. Thus, watching for a complete dissolution of the initial precipitate is the key for successful preparation of this polymorph. After all required amount of AgNO_3_ was added, transparent yellow solution was left in the dark to cool off slowly within ~10 h. Nice yellow blocks of the yellow polymorph of Ag(PiCO) appear in abundance. Yield 85%; complex rapidly decomposes at 160–162 °C. Analytical data: for AgC_7_H_9_N_2_O_2_ calculated, (found): C – 32.21 (31.81); H – 3.48 (3.51); N – 10.73 (10.62).

Those polymorphs showed different colors (Appendix A), as well as different solubility in water and organic solvents.

### 2.9. Silver(I) α -Oximido-(2-benzoimidazolyl)acetonitrile, Ag(BIHCO)

This yellow-greenish complex was obtained upon room temperature reaction between in situ deprotonated ligand H(BIHCO) (0.250 g dissolved in 10 mL of acetonitrile) and KOH (0.065 g dissolved in 2 mL of water) at 1:1 stoichiometric ratio. Resulting solution was filtered from small turbidity and aqueous solution of AgNO_3_ (0.198 g in 5 mL) was added dropwise upon stirring. The Ag(BIHCO) precipitate that formed immediately, was filtered, washed with 10 mL of water and then dried in a desiccator charged with concentrated H_2_SO_4_. Yield: 90%. Analytical data: for AgC_10_H_5_N_4_O_2_ calculated, (found): C – 37.41 (37.62); H – 1.57 (1.53); N – 17.45 (17.38). The complex is thermally stable to 150 °C.

### 2.10. Silver(I) α -Oximido-(1-cyano)acetonitrile, Ag(CCO)

This bright deep-yellow microcrystalline powder was obtained (pure) after precipitation of the [ONC(CN)2]- anion with AgNO_3_ from the reaction mixture containing CH2(CN)2 and HNO_2_ during the synthesis. Precipitated after the synthesis complex was filtered, washed twice with 5 mL of cold water and the dried in a vacuum desiccator over concentrated H_2_SO_4_. Yield: 96%. The Ag{ONC(CN)2} decomposes in the range of 175–210 °C. Thermally stable up to 140 °C without any visible change. Analytical data: for C3N3OAg calculated, (found): C – 17.84 (17.98); N – 20.81 (20.69).

The following twelve complexes have been prepared (listed with their commonly used abbreviations): Silver(I) Nitrosodicyanomethanide – Ag(CCO); Silver(I) α-Oximino-(acetamide)acetonitrile – Ag(ACO); Silver(I) α-Oximino-(ethylacetate)acetonitrile – Ag(ECO); Silver(I) α-Oximino-(benzoyl)acetonitrile – Ag(BCO); Silver(I) α-Oximino(pivaloyl) acetonitrile – Ag(PiCO), two polymorphs; Silver(I) α-Oximino-(N,N’-dimethylacetamide) acetonitrile – Ag(DCO); Silver(I) α-Oximino-(2-benzothiazolyl)acetonitrile – Ag(BTCO); Silver(I) α-Oximino-(2-benzoxazole) acetonitrile – Ag(BOCO); Silver(I) α-Oximino-(N-methyl-2-benzimidazole)acetonitrile – Ag(BIMCO); Silver(I) α-Oximino-(2-benzimidazole)acetonitrile – Ag(BIHCO); Silver(I) α-Oximino-(N-piperidine-acetamide)acetonitrile – Ag(PiPCO) and Silver(I) α-Oximino-(N-morpholyl-acetamide)acetonitrile – Ag(MCO). Yields, physical properties and data of elemental analyses for AgL are presented in Appendix A.

## 3. Physico-Chemical Characterization of Silver(I) Cyanoximates and Acrylate Polymeric Composited with Embedded Complexes

### 3.1. Characterization of AgL Using Spectroscopic Methods

Vibrational and electronic spectra were recorded for the initial protonated cyanoximes HL and for the final Ag(I) cyanoximates.

### 3.2. IR-Spectroscopy

Spectra were obtained from thick mulls in Nujol and were placed between two 15 mm diameter KBr disks using a Vertex 70 Bruker FT-IR spectrophotometer (Bruker, Billerica, MA, USA) set for 64 scans at 4 cm^−1^ resolution in the 4000–400 cm^−1^ range. Some neat samples powdery samples, however, were recorded using the ATR method (Bruker R32 spectrophotometer, Bruker, Billerica, MA, USA) when Nujol bands were obscuring bands of the CNO-fragment.

### 3.3. UV-Visible Spectroscopy

Spectra of diffused reflectance (SDR) were recorded from powders of solid AgL using Cary 100 Bio spectrophotometer (Agilent, Santa Clara, CA, USA) equipped with the Labsphere accessory (Appendix A).

### 3.4. Preparation of Polymeric Mixtures Containing Ag(I) Cyanoximates

After thorough drying in a vacuum desiccator, fine powders of Ag(I) cyanoximates were mixed with acrylate polymeric composite that has been modified for better flowability and handling to achieve a final concentration of 0.5, 1, 2.5 and 5% by weight (*w*/*w*) (Appendix A). Then 200 µL of each mixture was added to the wells of 96-well transparent flat-bottom plates (Corning Inc., NY, USA) using a syringe (Appendix A) and solidified using Spectrum curing light (Appendix A). Chlorhexidine®, a cationic polybiguanide (bisbiguanide)—commonly used to treat gingivitis and periodontitis—was applied at the same final concentrations and was used as a reference antimicrobial compound [49].

### 3.5. Solid Polymers Testing

Samples of acrylate-based polymeric composites were made as cylinders with ~1:1 ratio of length to diameter. The Ag(ACO) complex was arbitrary chosen for testing. A flovable polymer mixture (preparation of which was described above) was casted into 12 mm diameter and ~10–12 mm height cut small PVC tubes. Then a standard in dental practice 400 nm bright light source was used for the polymer curing. Samples were: (1) pure polymer, control; (2) polymer with 0.5 weigh % of the Ag-cyanoximate; (3) polymer mixed with 5% weight of the Ag-cyanoximate.

### 3.6. Mechanical Strength Testing

Strength of solid cylindrical samples of prepared polymers was measured using the instrument shown in Appendix A. Mechanical testing under quasistatic compression of pure polymeric composite and two samples with different amounts of embedded Ag-complexes (0.5% and 5% by weight) was conducted with an Instron 3369 universal testing machine at a strain rate of 2.5 mm/min, using a 50 kN load cell following testing procedures in the spirit of ASTM D1621−04a (Standard Test Method for Compressive Properties of Rigid Cellular Plastics). Results are presented in Table 1.

### 3.7. Bulk Density

These measurements were carried out in order to evaluate the effect of addition of silver(I) complexes on density of resulting polymer. Bulk densities (ρb) were calculated from the weight and the physical dimensions of the samples. Skeletal densities (ρs) were determined with helium pycnometry using a Micromeritics AccuPyc II 1340 instrument (Micromeritics, Norcross, GA, USA) depicted in Appendix A. Data are shown in Table 2.

### 3.8. Surface Area Measurements and Porosity of Composites

An attempt to determine the BET surface areas and possible pores sizes was made with an N_2_ sorption porosimetry at 77 K using a Micromeritics ASAP 2020 surface area and porosity analyzer (Micromeritics, Norcross, GA, USA), which is shown in Appendix A with results summarized in Table 2.

### 3.9. Leaching Experiments

Three Ag(I) complexes – Ag(CCO), Ag(ACO) and Ag(BTCO) were selected for testing Ag ions leaching because these cyanoximes represent a range of hydrophilic and hydrophobic ligands used to bind metal cations. These experiments consisted of five parts: (1) making bulk composite containing each of the selected three Silver(I) cyanoximates at 5% *w*/*w*; (2) casting them into a form of 3 × 10 mm; (3) curing the mixtures with light; (4) exposing the obtained solid pellets to non-halogenated buffers (TRIS and MOPS) at pH~7; (5) taking aliquots every 30 min for 170 h and analyzing their content for the presence of Ag ions using the ICP method. Further details of the experimental design of leaching can be found in Appendix A.

## 4. Biological Studies

### 4.1. Bacterial Strains and Growth Conditions

Four bacterial strains representing Gram-negative human pathogens *Pseudomonas aeruginosa* strains PAO1, initially isolated from a wound [50] and FRD1, mucoid cystic fibrosis isolate [51]) and Gram-positive *Staphylococcus aureus* strain NRS70, methicillin resistant respiratory isolate [52] and environmental isolate *Bacillus aryabhattai* SRP, were selected and tested. *P. aeruginosa* PAO1 and FRD1 are well-established lab strains originally received from Washington State University repository. *S. aureus* NRS70 was obtained from the NARSA repository. The identity of *B. aryabhattai* SRP isolate was confirmed by 16S rRNA sequencing followed by phylogenetic analyses. The cultures were grown in a 50% Brain Heart Infusion (BHI, for PAO1 and FRD1), 100% Tryptic soy broth (TSB) with 1% glucose (for NRS70) and 100% BHI with 1% glucose (for SRP) media. All the strains were grown at 37 °C.

### 4.2. Planktonic and Biofilm Growth Assessment

The 96-well plates pre-loaded with the mixtures of composite and Ag(I) cyanoximates or chlorhexidine were sterilized by exposing to UV light for 10 min. The sterilization quality was controlled by including a non-inoculated control. To minimize surface condensation and cross-contamination, the lids of the 96-well plates were treated with 0.1% Triton-100 and dried prior to the UV treatment. Both planktonic and biofilm modes of growth were quantitatively assayed. For this, prior to plate inoculation, the cultures of PAO1 and FRD1, NRS70 and SRP were grown in 50% BHI, 100% TSB with 1% glucose, 100% BHI with 1% glucose, respectively, to middle log phase as determined by the individual growth curves and diluted to obtain absorbance at 600 nm of 0.1. These normalized cultures were diluted 1:100 using saline (0.85% NaCl) and inoculated into the fresh corresponding medium in 96-well plates. The plates were incubated for 24 h at 37 °C. The media were replaced with fresh sterile media in 12 h of growth. To measure planktonic growth following the incubation, the cultures were transferred into fresh 96-well transparent flat-bottom plates and OD_600_ was measured using a Biotek Synergy HT microtiter plate reader (Tecan Instruments Inc., Mannedorf, Switzerland). The remaining surface-associated cells (biofilms) were washed three times with saline (0.85% NaCl) to remove loosely attached cells and stained with crystal violet (0.1%, *w*/*v*) for 10 min as described in Reference [53]. The dye was solubilized in 30% (*v*/*v*) acetic acid and 70% (*v*/*v*) ethanol and the absorbance of the extracted crystal violet stain was measured at 595 nm. Non-inoculated wells were measured as blanks and the corresponding measurements were subtracted from the rest of the data. Cells growing in the wells containing no composite or composites alone were used as positive controls. The data represent the mean values of seven technical replicates and two biological replicates. To consider batch-to-batch variability, every experiment was repeated at least two times and only the consistent data were considered for comparisons.

### 4.3. Scanning Electron Microscopy (SEM)

The inhibitory effect of Ag(PiCO)-yellow on four selected strains was studied by SEM. For this, PAO1, FRD1, NRS70 and SRP were grown on the surface of composite alone or the mixture of composite with 0.5–5% of Ag(PiCO)-yellow for 24 h at 37 °C. The planktonic cultures were removed and the surface-associated biofilms were rinsed with saline three times and fixed with 2% glutaraldehyde in 0.1 M sodium cacodylate, 9 mM CaCl_2_ for 2 h at room temperature. Following three washes with wash buffer (60 mM sodium cacodylate, 180 mM sucrose), the samples were fixed with 1% OsO_4_ (aqueous) for 1 h at room temperature, washed thrice again and dehydrated in ethanol (50%, 70%, 90%, 95% and 100% in 15 min steps). Then the samples were washed twice for 5 min each in hexamethyldisilaze (HMDS) and were air-dried. The plate was trimmed and Au-Pd coated. SEM imaging was done by using FEI Quanta 600 field emission SEM microscope available at the OSU Microscopy facility, with the settings at 15–20 kV and spot size of 3.0. Some wells in the plate showed the phenomenon known as ‘drift.’ Drift usually occurs when the sample is not adequately conductive. To overcome insufficient conductivity, we increased the thickness of the Au/Pd coating. However, some wells still showed drift to the extent that imaging was impossible. To enable imaging for these wells, we decreased the dwell time used in image capture from 30 microseconds to 3 microseconds.

### 4.4. A Combinatory Effect of Ag Compounds and Antibiotics

To test whether Ag compounds increase sensitivity to antibiotics, we tested Ag(PiCO)-yellow, Ag(PiCO)-red and Ag(BTCO) in combination with tobramycin or trimethoprim, the antibiotics commonly used to treat *P. aeruginosa* and *S. aureus* infections, respectively [54,55]. For this, we first determined the sub-inhibitory concentrations of antibiotics that cause reduction but not a complete inhibition of planktonic and biofilm growth of the pathogens. Then, 1% of overnight cultures with OD_600_ of 0.1 were added to 200 µL of the respective media containing antibiotics in 96-well plate (PAO1 and FRD1, 50% BHI with 0.5–6 µg/mL tobramycin; *S. aureus*, 100% TSB with 1% glucose and 0.3–2.5 µg/mL trimethoprim; and *B. aryabhattai*, 100% BHI with 1% glucose and 0.125–2 µg/mL kanamycin). Planktonic and biofilm growth was assayed as described above. Once, the sub-inhibitory concentrations were determined, the bacterial strains were grown in 96 well plates pre-loaded with the mixture of composite and selected Ag compounds and the corresponding medium: 50% BHI (*P. aeruginosa*) or 100% TSB with 1% glucose (*S. aureus*), which contained the corresponding antibiotics: tobramycin 0.5 µg/mL (*P. aeruginosa*), trimethoprim 1.25 µg/mL (*S. aureus*). Planktonic and biofilm growth in the presence of the silver compounds and antibiotics was monitored and analyzed as described above.

### 4.5. Durability Assessments

To determine the durability of the antibacterial effect of the Ag(PICO)-yellow, the cultures of PAO1 and FRD1, NRS70 and SRP were grown in 50% BHI, 100% TSB with 1% glucose, 100% BHI with 1% glucose, respectively and monitored for 6 days. The liquid cultures were collected for OD600 measurements every 12 h and replaced with sterile media. Biofilm growth was quantified at the end of 6-day incubation.

## 5. Results and Discussion

### 5.1. Chemical and Physical Studies

#### 5.1.1. Synthesis and Characterization

Considering the current utmost importance of developing new compounds with antimicrobial properties, here we synthesized and studied physical-chemical and antimicrobial properties of twelve silver(I) cyanoximates. Twelve solid pure Ag(I) cyanoximates (including two polymorphs obtained for the PiCO-system) of AgL composition (where L = cyanoximate anions from Scheme 1) were obtained in 83–98% yield (Appendix A). Prepared solid Ag(I) complexes represent colored microcrystalline powders (Figure 1). Their chemical composition was confirmed by elemental analyses for C, H, N content. Furthermore, the compounds were characterized by recording their solid state IR and electronic spectra (diffused reflectance) and data compared with those previously reported [43,47], thus confirming their identity. More specifically, the IR spectra evidenced the formation of coordination polymers in which the C-N-O fragment adopts the nitroso- character. Thus, typical for oximes ν(C=N) and ν(N-O) frequencies are no longer identifiable but two new ν^as^ and ν^s^ bands between 1100 and 1300 cm^−1^ can be observed [56] (Appendix A). This indicates the re-distribution of electron density in the fragment from the oxime- to nitroso-group that we observed previously.^32,36^ Electronic spectra of solid Ag(I) cyanoximates contain typical for nitroso- complexes *n*→π* transitions around 380–560 nm (Appendix A).

#### 5.1.2. Silver(I) Complexes: Their Structures and Preparation of the Polymeric Composites Mixtures

Crystal structures of several AgL we previously reported [43,47]. These compounds are mostly 2D-coordination polymers (with the exception of Ag(CCO), which forms 3D-network [47] containing nitroso-anions that form bridges of different complexity [42]. Also, we had determined crystal structures for both yellow and red polymorphs of the Ag(PiCO). They both crystallize in the same monoclinic crystal system but have different unit cell metrix and different donor atoms environments. Detailed description of their crystal structures is out of scope of this paper and will be published with polymorphs’ photophysical and spectroscopic properties in a separate paper elsewhere.

All the obtained silver complexes can be mechanically mixed with components of light-curable acrylate composite commonly used in dental practice. Both silver complexes and the polymeric composite tolerate each other remarkably well without any sign of chemical degradation. Moreover, silver(I) cyanoximates showed stability to visible light both as pure compounds and when mixed with the solid polymeric matrix. We empirically determined that the polymerization process has to be carried out using two different setups depending on the nature of the used silver(I) cyanoximate (Appendix A). Thus, when mostly *dark-yellow* or *redddish-pink colored* AgL complex was used for mixing with polymeric composite, the photo-polymerization reaction could be done using standard dental Spectrum curing light (Appendix A). On the contrary, when *light-yellow* AgL compounds (Appendix A) embedded into the light curable composite was used, the photopolymerization could be quickly achieved using an intense metal-halide lamp (Appendix A), where the 96-wells plate was placed directly onto the protective glass cover of the lamp. This resulted in the formation of a solidified composite within several minutes. No apparent signs of photodegradation of silver(I) cyanoximates and darkening were observed during the curing process. This is important finding since components of employed acrylate-based mixture (Appendix A) contains electron rich functional groups capable of chemical reduction of silver to the metallic state.

#### 5.1.3. Thermal Stability of Silver(I) Cyanoximates and Their Polymeric Composites

The effect of heat was estimated for pure dry silver(I) cyanoximates, pure Ag-free solid polymeric composite used as control and the polymeric composite containing 5% *w*/*w* of Ag(ACO), as an example. The final product of thermal decomposition of all tested AgL is metallic silver, which was easily observable as small shiny droplets inside an alumina crucible used in the thermal analyzer. The maximum temperatures, at which the individual AgL can be heated without decomposition are as follows: Ag(ACO) – 200 °C, Ag(BCO) – 140 °C, Ag(BTCO) – 160 °C, Ag(CCO) – 190 °C and Ag(ECO) – 190 °C. Overall, the obtained data clearly indicate a sugnificant thermal stability of the tested pure silver(I) complexes minimum to ~140 °C, with some samples being stable up to ~180 °C (Figure 2 and Figure 3). However, when complexes were embedded into the solid polymeric composite, there is slight decrease in thermal stability of the system (Appendix A). In any case, considerable thermal stability is important for future potential applications requiring thermal stability during heat sterilization (at 110 °C) of devices coated with compounds empedded into polymeric matrix.

#### 5.1.4. Cured Acrylate Polymers with Ag(I) Cyanoximates: Results from Mechanical and Porocity/Sorption Testing Data

All results of measurements presented below of the properties of the obtained solid acrylate polymers with embedded silver compounds represent at this initial stage of the project no more than proof-of-concept data. At first, we performed these testing with the aim to evaluate an effect of the presence of inorganic coordination polymer properties of a bulk organic polymeric composite in general. Therefore, a brief description of our basic finding is presented below.

The incorporation of silver complexes had rather negative effect on mechanical properties of the obtained solid polymers (Table 1). Thus, “dilutuon” of the polymeric composite mixture with siver(I) cyanoximate leads to roughly an order of magnitude lower values of both Young’s modulus and 40–100 times lowering the pressure at breaking point (Table 1).

Densities and porosities of solid light-cured acrylate composites containing Silver(I) cyanoximates were measured and averaged for two independent samples results presented in Table 2. All prepared materials turned out to be fairly compact with bulk and skeletal densities decreasing from pure solid polymer to one with 5% *w*/*w* of the compound to ~7.5% and ~2% respectively (Table 2). The effect of presence of inorganic polymer in solidified organic polymeric composite is evident. Thus, presence of metal complexes lowers the density of the polymeric material but increases its porosity. At this stage we attribute observed effects to inhomogeneity of the obtained solid polymers most likely associated with insufficient mixing procedures, rather than incompatible morphologies of Silver(I) coordination polymer and acrylate composite. This finding is valuable because low porosity and small density change of these new polymeric mixtures are necessary for the intended application as adhesives for indwelling medical devices.

Pure acrylate polymeric composites represent rather hard and practically non-porous, fairly dense materials. The introduction of silver(I) cyanoximate has led to the following changes:(1)Density of material became less;(2)Porosity of solid composite increases dramatically with addition of AgL;(3)Mechanical strength of material also significantly decreases.

Despite pronounced negative effects of “dilution” of a polymer composite with silver(I) cyanoximates on its mechanical properties and strength, there are still good possibilities for use of these antimicrobial compounds in medicine as adhesives for indwelling medical devices. The main objective is that there are no such large forces in living bodies that need to withstand MPa pressures and remaining intact. Real forces and pressures in living organisms are by order of magnitude smaller.

More detailed morphological and mechanical studies will be warranted and certainly conducted when a bigger picture of antimicrobial effect of polymeric composites containing our silver(I) compounds will emerge from studies that are currently underway.

#### 5.1.5. Leaching of Silver Ions from Solid Supports

It was very important to investigate whether solid polymeric composite with embedded inorganic Ag(I) coordination polymers are able to retain metal ions in the material. A significant presence of silver ions in aqueous solutions is very undesirable due to two main factors: (1) excessive amount is toxic; (2) removal of metal ions from the polymer will weaken its structure. Therefore, experiments on determination of free silver(I) ions in aqueous solutions after exposure of polymeric composites to water and buffers mimicking biological pH were warranted. To assess leaching of Ag^+^ ions, we selected three Ag cyanoximates: Ag(CCO), Ag(ACO) and Ag(BTCO) complexes that were incorporated into acrylamide polymer at 5% *w*/*w* concentrations as shown in (Appendix A). This choice was based on the hydrophilicity of the corresponding cyanoxime and the overall water solubility of the three AgL. The complexes Ag(CCO) and Ag(ACO) are sparingly soluble in water at elevated temperatures (>70 °C) but contain water soluble cyanoximes, while both Ag(BTCO) and its ligand precursor HBTCO are not soluble in water. Thus, the selected three cyanoximates cover a large spectrum of solubility and affinity to water for both components of the AgL: the ligand and the complex. As a control, a water-soluble silver(I) nitrate was used and incorporated into the solid matrix in a similar fashion. The concentration of Ag^+^ ions in solutions (two non-halogenated buffers TRIS and MOPS and in pure water) was monitored by the ICP method.

As expected, the control solid polymer containing water-soluble AgNO_3_ showed a significant leaching of Ag^+^ ions from its solid matrix (Appendix A). In this case, 32.1% of silver was released into water and 40.9% and 47.7% of silver leached into TRIS and MOPS buffers, respectively. To the contrary, leaching of Ag^+^ from the composites containing Ag(CCO), Ag(ACO) and Ag(BTCO) was below 0.2 ppm when monitored during over 150 h. Thus, the Ag(I) cyanoximates tested in the experiment did not to release any significant amounts of Ag^+^ from polymeric matrixes into the solutions of TRIS, MOPS buffers and DDI-water. These non-halogenated buffers had little effect on leaching of Ag^+^, presumable due to the significant thermodynamic stability of the starting coordination-polymeric Ag(I) cyanoximates. Indeed, all studied in this project AgL represent coordination polymers of different complexity, where the cyanoxime groups act as bridging ligands by means of the N- and O-atoms of the nitroso-groups and also using the CN-group as well. Nevertheless, monitoring of the leaching over the period of one week showed a very small leaching effect in the buffer solutions (both TRIS and MOPS) after the first 24 h. Further, with time there was no significant increase in leaching in all studied media – water and buffers (Figure 4 and Figure 5). It is interesting to note that, in the TRIS buffer, the leaching of Ag^+^ ions is ~30 times higher than that in the MOPS buffer solution at the same pH (Figure 4 and Figure 5). This result can be rationalized in terms of much greater propensity of TRIS molecule to act as a chelating agent (albeit week) rather than sulfonate MOPS molecule. In fact, a formation of the complex compound between Ag^+^ and TRIS was observed proving this common buffering reagent to be non-innocent with respect to metal ions in studied solutions [57]. Therefore, it has to be avoided by chemists in similar studies. In summary, the low level of Ag^+^ ions leaching from polymers containing Ag(I) cyanoximates can be explained by great thermodynamic stability of the studied AgL, which was reflected in their intrinsic poor water solubility.

### 5.2. Biological Studies

Prior to microbiological studies it was important to determine whether Silver(I) coordination polymers used in this work possess toxic properties for cells. Hence, we performed series in vitro cytotoxicity experiments to evaluate complexes’ effect on cells viability, which are summarized in Appendix A. No apparent cells toxicity was observed which allowed further investigations of solid polymers containing AgL for intended possible applications such as: (1) antimicrobial additives to adhesives for indwelling devices; (2) antimicrobial and biofilm inhibiting polymers for coating.

#### 5.2.1. Ag(I) Cyanoximates Inhibit Planktonic Growth of Gram-Positive and Gram-Negative Bacteria

To assess the bacteriostatic effect of the newly synthesized silver complexes, first we characterized the effect of all twelve compounds (Ag(ECO), Ag(ACO), Ag(BCO), Ag(CCO), Ag(BOCO), Ag(BiMCO), Ag(BiHCO), Ag(PiCO), Ag(MCO), Ag(PiPCO), Ag(BTCO) and Ag(PiCO) yellow and red forms) on planktonic growth of selected bacterial strains. We also included chlorhexidine, a cationic polybiguanide (bisbiguanide), commonly used to treat gingivitis and periodontitis, as a reference compound [49]. Three human pathogens, including two strains of *P. aeruginosa*, PAO1 (wound isolate) [50] and FRD1 (mucoid cystic fibrosis isolate) [51] and *S. aureus* strain NRS70 (methicillin resistant respiratory isolate) [52], representing different infection profiles, were selected for testing. Both *P. aeruginosa* and *S. aureus* are becoming increasingly resistant to currently available antibiotics [58,59]. In addition, we used an environmental isolate identified as *Bacillus aryabhattai* SRP based on sequencing of its PCR-amplified 16S rRNA gene (data not shown). Quantitative assessment of planktonic growth showed that eight compounds inhibited growth of all four tested organisms (Table 3).

The most efficient inhibition was observed against *P. aeruginosa* strains FRD1 and PAO1. Five compounds including Ag(BCO), Ag(PiCO), Ag(PiCO) yellow, Ag(MCO) and Ag(PiPCO) inhibited growth of at least one of the two strains at the lowest tested concentration of 0.5%. Two compounds Ag(ECO) and Ag(CCO) were similarly active against *S. aureus.* The isolate of *B. aryabhattai* SRP showed the overall highest resistance to the tested silver compounds.

The most significant inhibition of the SRP strain was registered at 1% for Ag(MCO) and Ag(PiPCO). Based on all four strains, we concluded that the most efficient compounds were Ag(PiCO), Ag(PiCO) yellow, Ag(MCO) and Ag(PiPCO). Overall, the cyanoximates showed the highest efficiency against *P. aeruginosa,* whereas *B. aryabhattai* showed more resistance. This may be due to the differences in the structure between Gram-negative and Gram-positive cell walls, elevated resistance of environmental bacteria and/or different modes of action of the tested cyanoximates.

Variability in the antimicrobial activity of silver compounds was anticipated based on our earlier report for Ag(MCO) and Ag(PiPCO) [45], as well as previously reported strain specificity of the antimicrobial effect of silver nanoparticles (AgNPs) [60]. The latter were tested against four strains of Gram-negative *Escherichia coli* and Gram-positive two strains of *B. subtilis* and three strains of *S. aureus*. However, in this case, *B. subtilis* strains showed the highest sensitivity for AgNPs, suggesting potential differences in the mechanisms of action between silver cyanoximates and AgNPs.

#### 5.2.2. Silver(I) Cyanoximates Inhibit Biofilm Growth of Gram Positive and Gram Negative Bacteria

It has been observed that silver effectively exerts antimicrobial effect against planktonic cells of diverse bacterial species [61,62] but is much less effective against biofilm cells [63]. Similar observations were made for AgNPs and ions tested against planktonic and biofilm cells of *P. aeruginosa* [64,65] and *Escherichia coli* [66]. Bacterial biofilms are well known to be more resistant than their planktonic counterparts. Once developed, biofilm infections are difficult to treat, mostly due to poor antibiotic penetration, slow growth, adaptive stress responses and the formation of persister cells. Further complication is the development of multi-species infections, where diverse pathogens express multiple different mechanisms of resistance. This makes it more difficult to devise the suitable interventions and the development of new anti-biofilm implant materials [67]. These highly resistant biofilm infections cost the healthcare industry more than $1 billion annually, as the only currently available effective treatment is a long-term high-dose antibiotic treatment along with removal of the infected device. To estimate the potential anti-biofilm effect of the synthesized silver complexes, we assessed their inhibitory effect on biofilm growth of the selected bacterial strains. It was expected that biofilms would show higher resistance to the compounds than the corresponding planktonic cultures. This was true for several cases, including Ag(MCO) and Ag(PiPCO), for which biofilms of at least three bacterial strains showed higher resistance (Table 4). However, the results for other compounds were strain specific, ranging from showing similar inhibitory effects against both planktonic and biofilm cells to being more efficient against biofilm cells. For example, Ag(BTCO) showed similar inhibitory effects against planktonic and biofilm cultures of *S. aureus* and *P. aeruginosa* but a significantly higher inhibitory effect (no growth at 0.5%) for *B. aryabhattai.*

Overall, the most stable inhibition against biofilms of all four bacteria was observed for both yellow and red forms of Ag(PiCO) inhibiting biofilm growth in the range of 0.5 to 2.5% (Table 4).

The apparent higher sensitivity of biofilms to some compounds is likely because the compounds were embedded into the substratum and prevented cell adhesion and therefore biofilm development as opposed to showing activity against developed biofilms. Interestingly, although chlorhexidine showed the highest efficiency against *S. aureus* and *B. aryabhattai*, it was not as efficient against biofilms of *P. aeruginosa* strains. Chlorhexidine, is a quaternary ammonium compound disrupting cell membrane and ultimately causing cell lysis and loss of cellular components [68]. The inability of chlorhexidine to inhibit *P. aeruginosa* strains is likely due to the up-regulation of RND-type multidrug efflux pumps together with down-regulation of genes encoding proteins involved in electron transport and membrane transport [69].

#### 5.2.3. Ag(PiCO)-Yellow Prevents Biofilm Formation Shown by Scanning Electron Microscopy (SEM)

In order to validate the inhibitory effect of Ag compounds on biofilm development, we performed scanning electron microscopy (SEM) of biofilms grown in the presence of Ag(PiCO)-yellow, one the most efficient compounds. For this, we applied three concentrations of the compound: 0.5%; 2.5%; or 5%, inhibiting biofilm development of, respectively; *P. aeruginosa, S. aureus* and *B. aryabhattai;* or all four strains. As a control, we observed the biofilm architecture and cell morphology in the samples grown on the surface of composite alone. All tested strains (*P. aeruginosa* PAO1 and FRD1, *S. aureus* and *B. aryabhattai*) formed uniform biofilms when grown on composite material with no Ag(PiCO)-yellow (Figure 6). In agreement with the biofilm growth experiments described above, the presence of 0.5% AgPiCO yellow practically abolished biofilm formation in *P. aeruginosa* strains PAO1 and FRD1. Only single cells or small cell clusters were detected on the treated surfaces. Similarly, the addition of 2.5% Ag(PiCO)- yellow prevented biofilm development of *S. aureus* and *B. aryabhattai* strains. The presence of 5% of the compounds completely prevented cell adhesion and no cells were observed on the treated surfaces (Figure 6).

It has been previously shown that silver ions destabilize the structures of the *S. epidermidis* biofilms likely by reducing the number of binding sites for hydrogen bonds and electrostatic and hydrophobic interactions [63]. Here we demonstrate that the biofilm development of two major antibiotic resistant pathogens *P. aeruginosa* and *S. aureus,* as well as environmental strain of *B. aryabhattai* can be significantly reduced or completely prevented by addition of 0.5 to 5% of at least one Ag(I) cyanoximate, Ag(PiCO)-yellow.

The antimicrobial activity of the studied compounds may rely on the release of Ag ions as was shown for other Ag-containing compounds [70,71]. It is likely that the newly synthesized cyanoximates serve as a vehicle delivering biologically-active Ag ions to bacterial cells. Further research is needed to study the interactions between selected cyanoximates and bacterial cells and human tissues. Development of more active compounds will further reduce the required dose and may enable their broad applications as additives to prosthetic glues or dental materials.

#### 5.2.4. Durability of Compounds

To assess the duration of the inhibiting effect of Ag compounds, we have performed long-term incubation of all four bacterial cultures in the presence of Ag(PiCO)-yellow. The inhibiting effect of 5% Ag(PiCO)-yellow for *S. aureus* and *B. aryabhattai* was stable for 36 h of incubation at 37 °C. However, it was significantly more extended for both PAO and FRD1 and lasted for 108 h. Even then, growth of *P. aeruginosa* strains reached only 10% of that of untreated cells. It took additional 12 h to recover the growth to the level of untreated cells. We also tested whether bacterial growth would re-occur on the surface of Ag(PiCO)-yellow-loaded composites after removing the previously grown culture by washing, sonicating and 97% ethanol treatment. However, sequential inoculations showed a minor increase in growth (data not shown). SEM analysis showed that the composites remained covered with the biomass, which likely shielded the antibacterial effect of Ag(PiCO)-yellow from the newly inoculated cells and provided support for their initial attachment and biofilm formation (data not shown). It is also possible that the remaining cells prevented the release of Ag ions and therefore diminished the antibacterial effect of the AgL. Further work will aim to further optimize the composition of the compounds that would further increase their antimicrobial properties and durability.

#### 5.2.5. Synergistic Effect of Ag(PiCO)-Red and Ag(BTCO) and Antibiotics

Since all available antibiotics are quickly becoming obsolete due to increasing bacterial resistance, there is a pressing need for combinatorial therapies, which would combine the application of antibiotics and the compounds sensitizing cells or preventing development of resistance, thus enabling efficient application of currently available antibiotics. Earlier studies suggested that Ag compounds have a synergistic effect with conventional antibiotics [70,72].

We tested whether the presence of selected Ag(I) cyanoximates would increase susceptibility to antibiotics of the studied pathogens (Figure 7). For this, we chose antibiotics that are commonly used against the respective bacteria and determined their sub-inhibitory concentrations. The latter were defined as the concentrations reducing but not fully inhibiting planktonic and biofilm growth of the corresponding bacterial strains. Based on planktonic and biofilm growth data, we estimated the sub-inhibitory concentrations of the antibiotics as follows: PAO1 and FRD1, 0.5 µg/mL tobramycin; *S. aureus*, 0.625 µg/mL trimethoprim; and *B. aryabhattai*, 0.125 µg/mL kanamycin. These concentrations were applied during growth of the bacterial strains in the presence of one of three silver compounds selected for this experiment, Ag(PiCO)-red, Ag(PiCO)-yellow and Ag(BTCO). Planktonic and biofilm growth was monitored and showed that the presence of Ag(PiCO)-red and Ag(BTCO) increased antibiotic susceptibility in both *P. aeruginosa* strains (Figure 7) but not in *S. aureus* and *B. aryabhattai* strains (data not shown). In FRD1, the combination of 0.5 µg/mL tobramycin and 0.5% Ag(PiCO)-red completely inhibited both planktonic and biofilm growth, whereas the presence of the antibiotic alone only slightly reduced growth. A similar trend was observed for 1% Ag(BTCO) in FRD1 and 0.5% Ag(BTCO) in PAO1. It was difficult to observe such effect at 1% Ag(PiCO)-red in PAO1, as almost no growth was observed at this condition. For the same reason, the effect Ag(PiCO)-yellow was not detected as it inhibited growth of PAO1 and FRD1 at 0.5%, the lowest tested condition (Table 3 and Table 4).

Combining Ag-based antimicrobials with other compounds with the aim to increase their efficacy have been previously proposed and tested in several studies. For example, combination of silver sulfadiazine with nitric oxide showed a synergistic effect against a wide range of bacterial pathogens causing wounds infection [73]. Storm *et al* used silver nitrate and nitric oxide combined together in a multilayered silica based xerogels and observed synergistic inhibition of *P. aeruginosa* and *S. aureus* attachment to the implant surface and enhanced the bacterial killing [74]. The presence of the ionic silver was also shown to enhance antibacterial properties of hydrogen peroxide against *P. aeruginosa* biofilms [75]. Chemically conjugated antibiotics with AgNPs enhanced the antibacterial effect of the antibiotics [76,77]. In our study, the increased sensitivity of *P. aeruginosa* to tobramycin, when applied in combination with Ag(PiCO)-red and Ag(BTCO), provides a potential for preventative and therapeutic treatment against *P. aeruginosa* biofilm infections. Using such combinatorial therapy will enable application of lower doses of antibiotics, which would reduce the overall toxicity and slow down the development of antibiotic resistance.

#### 5.2.6. Practicality of Studied Silver(I) Cyanoximates

The presented Ag complexes will bring three main benefits for future applications: (a) thermal stability of the pure complexes as well as their polymeric composites enabling their thermal sterilization; (b) pronounced antimicrobial activity of non-antibiotic origin allowing the compounds’ application as bactericidal agents alone or in combination with conventional antibiotics; (c) low metal ions leaching allowing the complexes to be embedded into light-curable acrylate polymeric matrixes for applications as coating materials.

## 6. Conclusions

Here we report the inhibiting effects of several silver cyanoximates against both Gram-positive and Gram-negative bacteria but particularly against two human pathogens, *P. aeruginosa* and *S. aureus*, suggesting that they can be used as efficient antimicrobials. In this initial “proof-of-concept study” detailed conclusions are as follows:Preparation of visible light insensitive, thermally stable and poorly water soluble Ag(I) cyanoximates presents a high yield one-step reaction at room temperature from respective cyanoximes and AgNO_3_ in aqueous solutions.The tested Ag(I) cyanoximates showed a significant thermal stability upon heating up to ~140 °C. Similar thermal stability was also found for the solid polymeric materials containing Ag(I) complexes, which is important for thermal sterilization purpose in their intended practical applications.The compounds demonstrate excellent tolerance to components of the light-curable polymeric composite and can be embedded into it at variable by weight concentrations from 0.5 to 5%. This is despite the presence of electron rich groups in ingredients of the composite mixture.Formed viscous, flowable polymeric composites containing AgL can be poured or casted into desired spaces and forms – well plates, plastic or rubber vials and vessels – and then successfully cured in them to a solid state by commercially available dental light source.The Ag(I) cyanoximates did not significantly leached out of polymeric matrices, which enables their applications in antimicrobial coating of surfaces.Introduction of solid silver complexes leads to the less dense final polymeric materials.Porosity of solid composites increases with addition of solid AgL into the polymer.Mechanical strength of cured solid acrylate composite significantly decreases when solid AgL are embedded into the polymer.The compounds showed significant antimicrobial effect against both planktonic and biofilm growth of Gram-positive and Gram-negative bacterial pathogenic and environmental strains. Importantly, selected compounds caused a synergistic effect when added together with conventional antibiotics. These results indicate a high potential for these compounds to be used as antimicrobials or as potentiates of antibiotics.

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
