# Peer review of "Polymeric Composites with Silver (I) Cyanoximates Inhibit Biofilm Formation of Gram-Positive and Gram-Negative Bacteria"

_polymers, 2019, doi:10.3390/polym11061018_

Round 1
Reviewer 1 Report
This paper by Lotlikar and coworkers described “Polymeric Composites with Silver (I) Cyanoximates Inhibit Biofilm Formation of Gram-positive and Gram-negative Bacteria”. The manuscript presents a new synthetic approach of silver/polymer composites that enhance bactericidal effect of currently available antibiotics. However, some additional information should be given and several issues need to be resolved before considering the manuscript suitable to be published.
The comments are as following:
1. The present structural analysis of polymer composites and other characterization results are quite weak and still not enough for an article in Polymers.
2. The dynamic mechanical properties, phase behavior and structure of polymer composites are essential for the comparison of performance with original polymer. These experimental data should be provided in the revised manuscript.
3. Detail information on morphology of neat silver(I) cyanoximates and polymer composites should be included. It is possible that the dimension and geometry of the silver(I) cyanoximates may affect their distribution in the composite matrices.
4. The results of tensile tests should be provided in the manuscript. In addition, authors should include the data and explanations for all the weight ratios. For example, it is more persuadable to include detail discussion of mechanical performance and biological properties on the silver(I) cyanoximate loading.
Author Response
We very much appreciate careful reading of our work by dedicated reviewers. Below
you may find our reply on your comments.

Reviewer 2 Report
The manuscript entitled “Polymeric Composites with Silver (I) Cyanoximates 2 Inhibit Biofilm Formation of Gram-positive and 3 Gram-negative Bacteria” demonstrated that comparison in between antimicrobial activity and biofilm inhibition of various silver(I) cyanoximates. The concept of the paper is fine; however, the research data to support results and conclusion is weak in the current form. The issues are listed as below.
Author must present strong evidence for their presented concept of biofilm inhibition and antimicrobial activity of silver cynoximates. Author must provide experimental results related surface marker, molecular mechanism and microscopic images for in-depth study (such as PCR, western blots and fluorescence images, immunochemistry images etc.)
Author must do toxicity evaluation on cells or cell lines. Toxicity of any physical and chemical agent must be evaluated on primary or normal cells for validation.
Quality of all figures are weak, author must improve quality of figures using professional software’s.
Author Response

(The authors gave the same response as above.)

Round 2
Reviewer 1 Report
Although this study has its own specific objectives and technical difficulties, the manuscript has slightly changed with providing some explanations and thus would be suitable for publication in its current form.
Reviewer 2 Report
I recommend to accept this manuscript as author have properly responded